# Ruff Morning? The Use of Environmental Enrichment during an Acute Stressor in Kenneled Shelter Dogs

**DOI:** 10.3390/ani13091506

**Published:** 2023-04-28

**Authors:** Pamela Dare, Rosemary Strasser

**Affiliations:** Neuroscience and Behavior, Psychology Department, University of Nebraska Omaha, Omaha, NE 68182, USA; rstrasser@unomaha.edu

**Keywords:** enrichment practices, stress reduction, captive animal welfare, sensory stimulation, shelter dogs

## Abstract

**Simple Summary:**

Dogs in shelters are often exposed to unavoidable stressful events. Finding effective and novel enrichment for dogs, especially the longer they stay in that environment, is a significant welfare concern. We wanted to compare the effectiveness of enrichment items as a stress buffer for shelter dogs during the acute stressor of the morning cleaning and to support a proposed prescription system of enrichment. This new system would focus on post-enrichment behaviors. We found calming items were more effective than no enrichment at reducing stress-related behaviors seen in kenneled dogs. Tactile items reduced vocalization the most, with the arousing tactile item (a ball) resulting in the lowest frequency overall. Our results suggest that the type of enrichment item should be thoughtfully chosen during acute stress events, which would then optimize a shelter’s limited resources, decrease stress-indicative behaviors, and indirectly reduce the need for euthanasia by increasing adoption rates.

**Abstract:**

The length of stay for some animals has increased with the recent reduction of animals euthanized in US shelters and animal control facilities. Research examining the effectiveness of different types of enrichment on buffering the effects of acute daily stressors in the shelter environment, such as kennel cleaning, is lacking. In addition, daily known stressors can result in undesirable behaviors that could lead to a need for euthanasia. Ways to effectively reduce the effects of daily stressors while optimizing strained resources is currently a high priority. In this study, we presented shelter dogs with food, tactile, and scent enrichment items to increase (arousing) or decrease (calming) activity during the daily stressor of morning kennel cleaning. We found that calming, rather than arousing, enrichment items were associated with body position scores indicative of lower stress in dogs, with calming scent enrichment (lavender) producing the most significant benefit. In contrast, items that showed the greatest reduction in vocalization were arousing (ball) compared to other arousing conditions. Our findings suggest that different unwanted behaviors in the kennel environment often associated with stress can be reduced using specific types of enrichment during a daily stressful event. Further, the results illustrate that enrichment items other than food might be more effective at decreasing certain undesirable behaviors. Overall, this study provides insight into how shelter workers might effectively use enrichment items during an unavoidable acute stressor. With many shelters keeping dogs longer, addressing events that might cause repeated stress in this population may indirectly help with adopting and lowering euthanasia rates due to unwanted behavior that develops due to repeated exposure to this necessary but acute stressor of morning cleaning.

## 1. Introduction

At least 3.1 million dogs enter shelters annually in the United States, and just under 400,000 dogs were euthanized in 2019 [1]. Decreasing the euthanasia rate while providing each of these dogs with proper care and food requires substantial shelter resources. In addition, the average length of stay for dogs in shelters historically tends to rise as euthanasia decreases [2], and a longer length of stay means each dog is using more shelter resources, on average, compared to the past. The increased length of stay is also a priority welfare issue considering the dogs will be undergoing potential stress for long periods. Even if animals have the best physical care possible, they can still experience poor welfare if their mental requirements are unmet [3].

Regardless of the minimum level of welfare provided, there are particular (possibly negative) situations no organization can realistically avoid, such as surgery and general husbandry procedures. Despite their necessity, the captive animal can perceive these events as negative. In fish, for example, rigorous tank cleaning lowered survival rate and decreased their immune response compared to less invasive scrubbing [4,5]. Minimized immune responses leave animals vulnerable to sickness and may slow down their recovery, potentially leading to early euthanasia in an attempt to save resources for healthier animals. In shelters especially, cleaning procedures cannot be avoided because of the average dog’s urination frequency. These practices also cannot be greatly altered, such as not using power washers, due to the need for efficiency in overcrowded and underfunded shelters. Though shelter dogs’ responses to husbandry practices have not been investigated, we do know in several other species of mammals (such as rats and marmosets), practices such as restraint and weight checks, were shown to impact physiological variables and stress-related behaviors negatively, leading to a decrease in their overall welfare [6,7,8,9,10,11]. The only solution remaining is to research easy modifications to standard operating procedures to increase the overall welfare of the dogs.

### 1.1. Behavioral Welfare Measures in Captive Animals

There are different measurements that researchers can use to try to evaluate the welfare of their animal subjects. Behavior measurements are used because they are cost-effective and have been used in numerous studies to determine the animal’s welfare state in question [12]. Observing the animal’s behavior may be one of the best indications of preference or aversion [13]. It can indicate different emotional states [14]. Shelter staff can then use picture representations of correlated behaviors to determine the animal’s overall stress level. These representations often include overall ‘body position’, body language, and other validated stress behaviors, which can be used to assign a stress score or value [15]. For example, common behavioral indicators of poor canine welfare include frequent vocalizations, crouching (i.e., heads below the chest line), flattened ears, or a lowered tail [2,15,16,17,18].

One of the things welfare evaluations indirectly measure is an animal’s stress level. There are two types of stress, chronic (commonly studied) and acute. The latter includes short periods where an organism undergoes higher than normal stress levels, such as being restrained. Loud and/or unpredictable daily cleaning would be a typical example of an activity that results in acute stress for a kenneled dog. With this example, the dog undergoes at least one example of acute stress (i.e., daily cleaning) while dealing with the chronic stress of being in a new environment (the shelter). Because chronic and acute stress has adverse health effects on animals [18], understanding the effect of husbandry on stress levels is a priority welfare concern.

Shelter dogs experience both acute and chronic stress in varying degrees [18], which can lead to immune suppression in dogs [19] and illness in cats [20]. This suppression reduces the animal’s ability to fight off infections, which can further drain shelter resources. Noise levels in shelters (usually over 100 dB) are above the OSHA regulations for factory workers [21] and are one example of a chronic physical stressor that dogs cannot avoid. Another potential chronic stressor (for a social species such as dogs) is the standard shelter setup of single housing, which is associated with erratic movements and increased barking [22]. A common source of acute stress for shelter dogs is being newly admitted. This change in environment usually results in a characteristic cortisol spike for at least three days after initial admittance [19] and has been shown to lead to a heightened immune response [2]. Unlike the impact of being newly admitted, the effect of the acute stress resulting from daily cleaning on shelter dogs has not been well studied.

### 1.2. Environmental Enrichment and Stress

Chronic stress in captive animals has been well studied. The most common method used to reduce chronic stress is adding items to the environment, referred to as enrichment, which has decreased the frequency of certain behavioral disorders [22]. Enrichment is used in shelters and has been shown to improve the overall welfare of dogs [14,21,23,24,25]. Enrichment plans are successful if they increase desirable behaviors, such as play, and decrease stereotypical behaviors or other behaviors associated with poor welfare, such as frequent vocalization [26]. Stereotypical behaviors are defined as relatively repetitive behaviors that seem to have no immediate purpose and may be a means of coping with poor welfare, either in the past or present environment [2,27]. It has been found that dogs with an enrichment program were significantly more likely to pass their behavioral test [28].

Some have categorized environmental enrichment into animate and inanimate forms, such as inanimate types of enrichment focusing on affecting different senses, such as food, scent, and tactile enrichment, including toys and blankets [29,30]. However, further classification or comparisons are currently lacking, and in practice, most organizations give out enrichment with the belief that all enrichment is equally beneficial in all situations. Despite the amount of research conducted on specific enrichment items and human contact in the shelter [14,16,18,24,26,30,31,32,33,34,35,36], the effectiveness of different types of enrichment items in reducing stress during morning cleaning is one area yet to be explored.

The categories of enrichment could be further explored based on whether the item is more calming or arousing to the animal. Calming enrichment would consist of items meant to focus the animal on a specific item. Arousing enrichment would include items that are meant to encourage the energetic mental stimulation of the dog by an item in their kennel. Rather than organizations focusing on each behavior in isolation (i.e., jumping, fly snapping, or barking), focusing on groupings of behaviors could be more helpful in decreasing the proportion of overall undesirable behavior in one situation. Calming enrichment may be effective for overactive/reactive dogs, which often display multiple negative behaviors such as wall jumping and a high barking frequency. Arousing enrichment will be more beneficial for dogs with behaviors on the other extreme, such as fear, aggression, or hiding. Investigating which enrichment items could be optimal for individual dogs or behavior problems would enhance their effectiveness as stress reducers.

Beyond the kind of enrichment items mentioned above, enrichment that can target different sensory systems may also change the behavior of animals in different ways. Food enrichment (i.e., anything that involves access to food, such as puzzle boxes) is found in almost half of all studies with zoo animals [27]. Scent enrichment is one of the newest forms of enrichment for all animals [18,30], with most of the work done with cats and felids [29]. Certain scent items have been found to have a calming effect on dogs [29]. For example, lavender has been shown to encourage calm behaviors and decrease vocalization frequency during the acute stress of care rides [34,37,38]. In contrast, few studies have examined arousing scents. Prey urine was also found to increase overall activity levels, suggesting that arousing scent enrichment also influences behavior [38]. Overall, the effect of different scents on canid behavior is not well studied [39] and therefore requires more investigation to better our understanding. Tactile enrichment, such as blankets or toys, is one of the most common forms of enrichment used in shelters; however, despite its prevalence, it has not been well studied [29]. Previous studies have suggested looking more specifically at the type of toy (i.e., hanging versus laying on the floor) presented to dogs [40] instead of treating all toys equally enticing. This suggestion is important because toys that the dog ignores illicit no interaction and therefore are not effective enrichment items. Although different types of enrichments could engage different sensory modalities, they have not been compared in a single study to determine their effectiveness as a stress buffer during morning cleaning.

### 1.3. This Study

We investigated different types of enrichment during an acute unavoidable stressor (morning cleaning) at a local humane society. We measured body position and vocalization rate, as they are known indicators of stress or negative welfare in shelter dogs [15,26,34,38].

The first aim of this study was to compare how effectively different sensory types of enrichment reduce (buffer) behavioral measures of stress. We hypothesized that any enrichment would reduce vocalizations and produce body positions indicative of lower stress [29,34,38]. Given the current prevalence of food enrichment items [27], we predicted that food would result in fewer behavioral indicators of stress than scent or tactile.

The second aim of this study was to determine if one of the proposed categories of enrichment (calming vs. arousing enrichment) is more effective at reducing negative, stress-induced behaviors. For this aim, we hypothesized that calming enrichment would lower vocalization frequency and body position score compared to arousing enrichment.

Although previous studies have examined some of these variables in captive animals individually, it has been suggested that this only provides one side of the picture [40]. Therefore, examining multiple forms of enrichment in one environment may help determine which enrichment type may be most beneficial to the captive animals.

## 2. Materials and Methods

### 2.1. Subjects

All dogs were housed at the Nebraska Humane Society (N = 83). Exclusion criteria included dogs not yet legally considered the shelter’s property or had known behavioral/medical holds. The average age was 40.55 months old (SD = 31.64 months). The average weight was 25.15 Kg (SD = 12.30 Kg). There were 32 females, 50 males, and 1 unknown sex that was not recorded on the kennel paperwork at the time of testing.

### 2.2. Kennel Design

Kennels were arranged in two rows of 10 with a walkway between the rows and walkways bracketing the outside of the kennels. They had a ‘front’ portion 10 × 4 feet with four feet high walls. The front was separated from the ‘back’ portion by a solid upper wall and a lower metal sheet that could rise and fall to lock the dog on either side. The back portion was 3 × 4 feet with four feet high walls. All kennels had blankets (some also have a raised plastic bed), a food bowl, a water bowl, and at least one toy in the front portion. No items were in the back portion, and studies have shown that a basic setting is not physically or mentally arousing [36]. The dogs were kept in the front during the day with the metal sheet down and limited or no access to the back portion of the kennel except during cleaning. Dogs were observed at a roughly 45-degree angle from the kennel against the wall (3-foot clearance) in their shelter-assigned kennel and not moved for this study. Researchers were standing and facing the dogs to be able to record vocalization as it occurred (ensuring the vocalization was from the dog under observation and not a neighbor). Figure 1 shows a summary of a standard kennel under observation.

### 2.3. Treatment Conditions

There were six enrichment conditions, with one control condition (no enrichment). These conditions were split into two categories of enrichment that were intended to simulate either arousing or calming behaviors in dogs. The enrichment also varied by sensory mode: food items, unfamiliar scents, or tactile items. Shelter volunteers premade all food items. Arousing food items were toilet paper tubes that were pre-filled with food (either peanut butter, wet dog food, or pumpkin), requiring dogs to tear them apart to eat the food. Calming food items were dog size appropriate KONG^®^s stuffed with peanut butter, which resulted in dogs lying down and licking to consume the food. The focus of the food items was on the extraction process rather than the type of food used, which was often similar between the calming and arousing items. Arousing scent items were white dish rags with one milliliter of rabbit urine [38] diluted in two cups water: 4 fluid oz urine concentration. Calming scent items were white dish rags with one milliliter of lavender essential oil [29] diluted in 2 cups water: 20 drop oil concentration. Dilutions were made according to bottle suggestions for effective scent diffusion. These rags were placed on the floor in the kennel’s corner. Arousing tactile items were balls, either tennis or Nerf. Calming comfort items were fleece blankets placed in the back part of the kennel. Figure 2 summarizes the conditions.

Enrichment conditions were preassigned using Latin Square Random Assignment based on the day of observation. Each day was randomly assigned a sensory mode, and the category was counterbalanced within that day. The only exception were dogs that already experienced both types of enrichment, as repeat conditions were unwanted. These dogs were randomly assigned to another condition as needed to counterbalance the sample size per condition.

### 2.4. Study Procedure

Data was collected once the morning cleaning started for the day after all dogs were given food and time to eat (approximately an hour after they were given food). Researchers had no contact with the dogs, enrichment items were placed in the back before the dogs were placed there by shelter workers per their cleaning protocol. All variables (body position and vocalizations) were collected in two-minute rounds while the dog was in the back. Most cleanings occurred while the dog was in the kennel’s bare and smaller back area. A round started with vocalizations being recorded on an all-occurrence basis per interval and ended with recording the overall body position. As mentioned above body language, and other validated stress behaviors can be used to assign a stress score or value [15]. Body position score figures were used previously in our lab and the shelter under study based and were compiled from sources [41,42]. Researchers underwent a testing procedure before live scoring to ensure the pictures were applied consistently, with high interrater reliability. Figure 3 details the standard body position scale that was used.

Each observer had multiple dogs per observation day. The dogs were observed for one round at a time, following the same order (i.e., dog 1, dog 2, dog 3, dog 1, etc.), to get equal observation time per dog. Intervals between the observation periods differed based on the number of dogs available for testing each day but stayed below 30 min. This time frame was equivalent to a continuous sample [43,44]. For a dog’s data to be included in data analysis, it was required to have at least two rounds of data per observation day.

Observations were completed using live scoring by a team of trained assistants. For inter-rater reliability, one dog a day was recorded with a video camera for scoring. Each researcher was trained on the behavior definitions and body position figures and quizzed for accuracy. Raters had to obtain an 80% on the quiz to advance in training. Following this, they were also required to shadow the lead researcher. Inter-rater reliability of the videos was scored by the trained assistants and compared with the lead ‘researchers’ answers (*r* > 0.8, *p* < 0.05). When collecting data, the ethograms given to the researchers were organized based on active/inactive behaviors to remove researcher bias of categorizing stereotypical behaviors.

### 2.5. Data Analysis

The main effects of enrichment presence, category (arousing vs. calming), and sensory modes (food vs. scent vs. tactile) on vocalization frequency and body position were analyzed using SPSS and the mixed measures program. This program runs an analysis similar to an ANOVA with multiple control variables/covariates, and all t-tests comparing the dependent variable means of the groups. Our first set of ANOVAs was a one-way ANOVA comparing the presence of enrichment to no enrichment for both body position and vocalization separately. Then, we split enrichment based on the proposed arousing vs. calming descriptions and ran a one-way ANOVA comparing those groups for both dependent variables. Next, we ran a one-way ANOVA comparing the three sensory modalities (food, scent, and tactile) as well as the no enrichment condition, again for body position and vocalization. Lastly, we ran a 2 × 3 ANOVA comparing (calming vs. arousing) and (food, scent, tactile) interactions on both dependent variables. All results were analyzed controlling for sex, source, temperament, length of stay, age, weight, and the order in which the enrichment was given. All condition means were estimated with condition order covariate value of two. Only significant results are reported at the α < 0.05 level. On all stress or fear measurements, lower numbers indicated a higher welfare state; therefore, decreased, or reduced measurements reflect the desired welfare outcome.

## 3. Results

ANOVAs and means of the main effects are summarized in Table 1.

We first investigated whether the presence of enrichment was significantly different from no enrichment during acute stress. As seen in Figure 4, enrichment significantly lowered vocalization frequency compared to the no enrichment condition (F(1,943) = 9.55, *p* = 0.002). We also examined if there was a difference between calming and arousing enrichment during acute stress (see Figure 5). On days when calming enrichment was given, a lower body position score associated with a more relaxed state was observed (F(2,862) = 3.64, *p* = 0.027) compared to when arousing enrichment was presented. The last main effect we examined was whether one of the different sensory modality enrichments—food, tactile, or scent items—was more effective at the group level (Figure 6). When given scent enrichment, dogs exhibited the most relaxed body position (F(3,862) = 2.72, *p* = 0.043). In contrast, during days with tactile items, dogs had the lowest vocalization frequency (F(3,862) = 4.00, *p* = 0.008) compared to the other sensory items.

In addition to the main effects described above, we also investigated if there were any interactions between the categories (calming vs. arousing) of enrichment and the different sensory modalities (food, scent, tactile) summarized in Figure 7. In the calming enrichment category, scent items produced a lower body position than food or tactile items (F(2,862) = 4.05, *p* = 0.018). When comparing the two scents (lavender vs. rabbit urine), the body position score was lowest with the calming (lavender) scent (F(1,862) = 5.73, *p* = 0.017), indicating that the calming scent produces the most relaxed state.

In contrast, there was a difference in vocalization frequency within arousing items. When tactile enrichment was given, dogs exhibited fewer vocalizations than when presented with food or scent items (F(2,881) = 6.30, *p* = 0.002). However, between the two food items (KONG^®^ vs. tube), calming enrichment (KONG^®^) resulted in a lower number of vocalizations than the arousing item (F(1,881) = 3.85, *p* = 0.050).

## 4. Discussion

In this study, we examined whether certain types of enrichment would induce a relaxed state in shelter dogs, indicated by a decrease in vocalization frequency or lower body position scores indicating a relaxed state, during the unavoidable acute stressor of morning cleaning. As expected, enrichment during morning cleaning decreased the frequency of vocalizations compared to when there was no enrichment. In addition, we found evidence that different types of enrichment were more effective than others at lowering vocalization frequency and improving body position scores associated with a less stressful state.

As predicted, enrichment during morning cleaning produced a 31% decrease in the frequency of vocalizations from dogs. Previous research has shown that increased vocalization frequency is correlated with a decrease in overall welfare [22,26]. Noise levels increase significantly during cleaning times, so reducing vocalization frequency would also help reduce this welfare issue [45]. Therefore, this study supports the idea that enrichment during the acute stress of morning cleaning, which has not been measured before, increases shelter dogs’ welfare. Additionally, decreased vocalizations will reduce overall noise levels during that acute stressor, benefiting all dogs in the same kennel room. Shelters often have sustained high levels of noise, which may physiologically compromise the dogs [45]. As multiple studies have reported the benefits of enrichment during chronic stress [14,23,24,25,33], our study results also emphasize the importance of incorporating enrichment items during acute stress moments in shelters.

In this study, we provided calming items to focus a dog’s attention or arousing enrichment objects to release energy through stimulation or reduce boredom. Given the acute stress of morning cleaning, it was hypothesized that calming enrichment would be more beneficial than arousing items for reducing negative arousal. A more relaxed body position score supported this hypothesis, and an overall decrease in vocalizations during calming conditions compared to arousing enrichment or no enrichment days. We acknowledge that the intent of the enrichment, to calm or arouse, is based on behavioral interaction with the items and may differ between individual dogs. Future studies should examine other variables, such as arousal of the HPA axis and differences in individual responses to these different enrichment items.

Though studies have not addressed calming versus arousing types of enrichment before, some have applied the overall concept, such as when lavender decreased vocalizations in car rides [34,38]. Another study by [35] found significant differences in cortisol levels between dogs with female and male petters (potentially tactile stimulation). This study also found that despite superficial similarities in the petting technique, dogs with male petters had higher cortisol levels than dogs with female petters [35]. In a follow-up study, once males imitated the soothing and calming tone and style of female petters, this difference in cortisol level became insignificant [35]. This result shows that calming behaviors during stressful situations are likely to reduce negative arousal states, consistent with the hypothesis in this study. Another experiment found that increasing or introducing daily walks (an activity that releases energy) reduced stereotypies in dogs in a home setting [46]. These studies, combined with the results from our study, suggest that dogs, on average, respond differently to calming and arousing enrichment items. Future studies, however, should further examine individual differences among dogs in response to these different enrichment types. Other factors that could influence the effectiveness of enrichment items include a dog’s temperament and length of stay at the shelter [47].

Although most shelters use food enrichment, recent research on other sensory items suggests that enrichment protocols may be diversifying [29,34,38,39,40]. However, few studies have directly compared sensory types (i.e., food, tactile, scent) of enrichment in shelter dogs. Due to the popularity of food enrichment, we hypothesized that food would be the most effective at reducing the vocalization and body position associated with stress. Overall, this hypothesis was not supported. This result was surprising due to the large proportion of food enrichment given daily in captive environments [27]. We found that dogs displayed a more relaxed body position when given scent enrichment, and vocalization frequency was lowest when given tactile enrichment. Notably, neither of these common shelter behavior issues decreased with food enrichment. This result further supports the concept that the type of enrichment item should be thoughtfully chosen during acute stress according to specific behavioral deficits or observed issues. It also supports the growing base of literature [29,34,38,39,40,41] that including diverse types of enrichment is necessary and appropriate for the long-term welfare and psychological needs of captive animals.

Due to the lack of past research on sensory enrichment in shelter dogs, we did not posit a hypothesis for the interaction of type of enrichment (calming vs. arousing). We did see significant differences between the calming and arousing conditions, with calming scent (lavender) enrichment producing the most relaxed state and calming food (a KONG^®^) producing the fewest vocalizations. Previous studies [2,15,16,17,18] have shown a more relaxed body position and fewer vocalizations are indicators of low stress in kenneled shelter dogs. Therefore, these findings further support the idea that calming enrichment items, more than arousing items, can mitigate the stress dogs experience in shelters and increase overall welfare, suggesting a need to prescribe enrichment to specific situations.

This study is not without its limitations. We could not control every variable, such as whether the dog was adopted or moved to a different kennel area during the duration of the study. Because of this lack of control, we only examined group differences, and future studies should investigate individual differences. We also could not control the time a worker spent cleaning daily, meaning we could not always complete a full set of rounds with each dog. In response to this issue, we set a minimum number of completed intervals per dog for inclusion in the study.

## 5. Conclusions

Cleaning in shelters is unavoidable. Unfortunately, this cleaning also causes acute stress for shelter animals. In dogs, this stress has been shown to lead to overactivation of the HPA axis and may be correlated with developing diseases such as arthritis, diabetes, and cancer (see 48 for a review) [48]. However, this welfare issue can be improved relatively easily by the use of specific treatment enrichment items. Although we did not examine euthanasia rates in this study, future studies could examine if reducing unwanted behaviors during acute stress events can indirectly reduce euthanasia rates.

Although enrichment in shelter dogs has been examined before, this is the first study to compare different sensory enrichment items intended to calm or stimulate a dog during a stressor. The investigation into whether enrichment can mitigate the adverse effects of unavoidable husbandry will help shelters improve the overall welfare of the animals. Additionally, comparing how different types of enrichment buffers acute stress can allow shelters and other institutions to use their resources more efficiently. It is time to re-evaluate the proper strategy for implementing enrichment in these environments.

## Figures and Tables

**Figure 1 animals-13-01506-f001:**
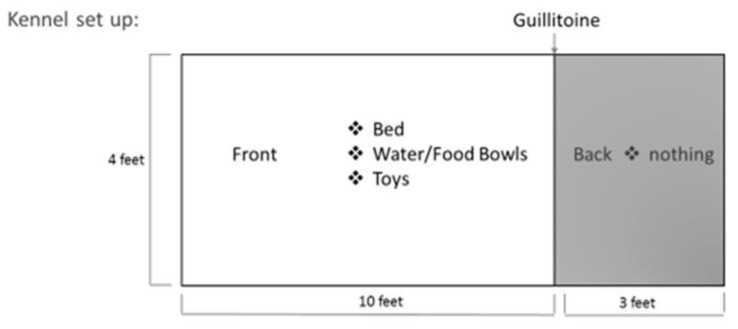
A summary of a standard dog kennel at Nebraska Humane Society.

**Figure 2 animals-13-01506-f002:**
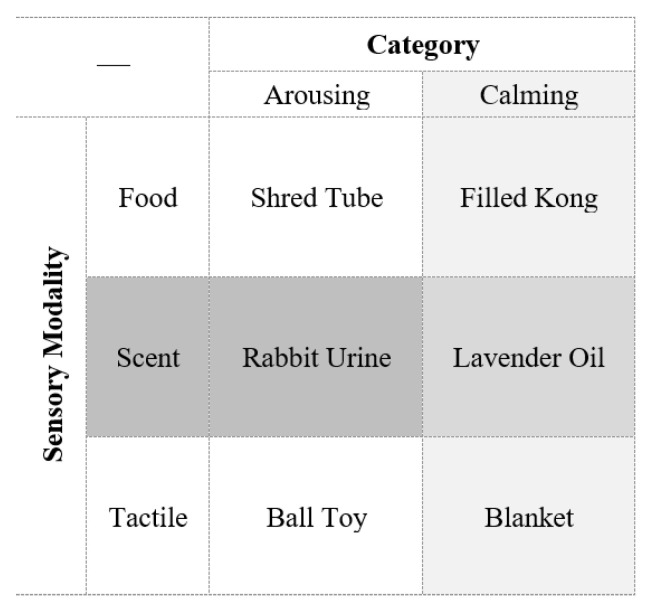
The experimental enrichment condition breakdown. Not shown is the control condition, which had no enrichment added.

**Figure 3 animals-13-01506-f003:**
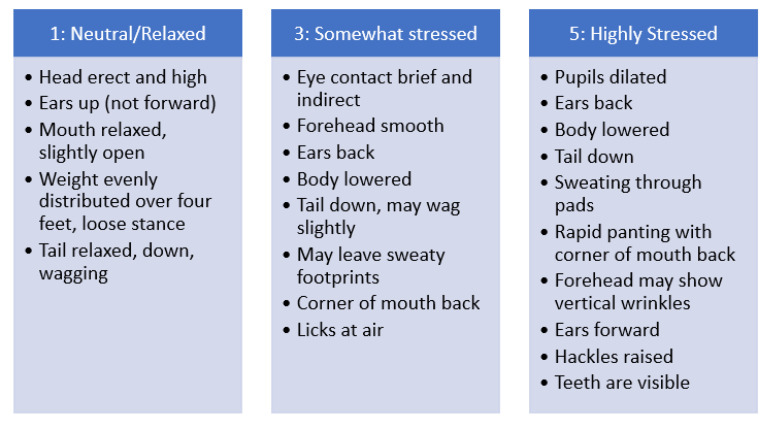
Standardized body positions are used to rate overall body position at the end of the round. Scores of two or four were also allowed if the body position was a mix of the three listed values. Written descriptions are given in the figure above but illustrations used can be found at [41,42].

**Figure 4 animals-13-01506-f004:**
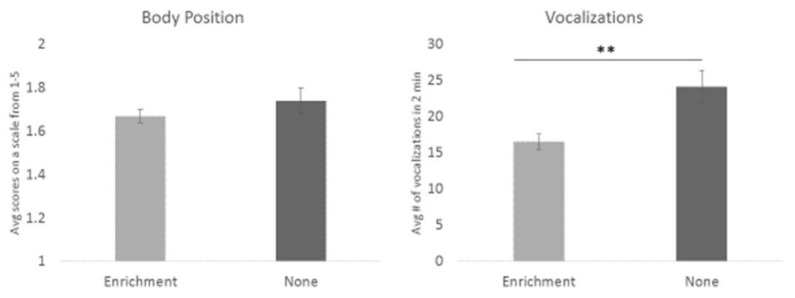
Comparison of the means of body position and vocalization frequency for enrichment presence to no enrichment. The presence of enrichment significantly affected vocalization frequency, such that dogs vocalized less on average when given enrichment. Asterisks denote significance below the *p* < 0.01 level. Error bars show ± one standard error.

**Figure 5 animals-13-01506-f005:**
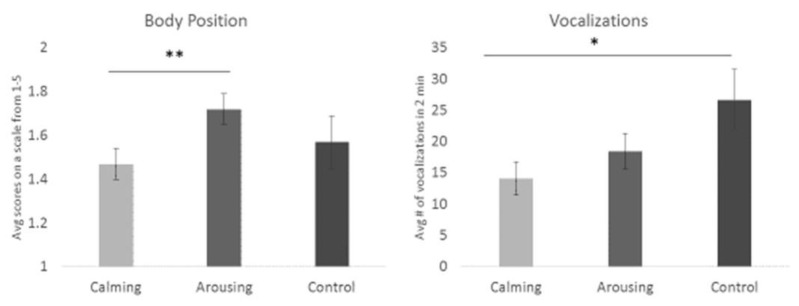
Comparison of the means of body position and vocalization frequency for calming to arousing enrichment. Body position score was lower (i.e., more relaxed) when dogs were presented with calming enrichment than arousing items. Calming enrichment items produced a significantly lower vocalization frequency than no enrichment. One asterisk denotes significance below the *p* < 0.05 level. Two asterisks denote significance below the *p* < 0.01 level. Error bars show ± one standard error.

**Figure 6 animals-13-01506-f006:**
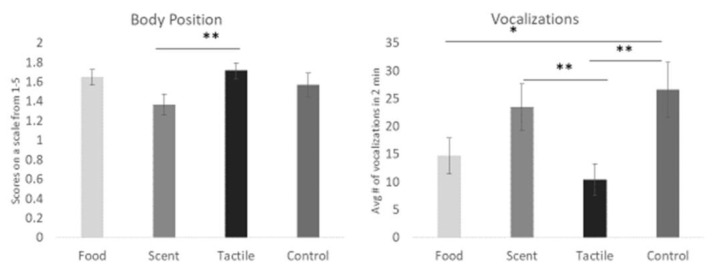
Comparison of the means of body position and vocalization frequency for food to scent to tactile enrichment. Scent items resulted in a lower body position score (i.e., more relaxed) than tactile items. The tactile items produced significantly fewer vocalizations than scent and no enrichment One asterisk denotes significance below the *p* < 0.05 level. Two asterisks denote significance below the *p* < 0.01 level. Error bars show ± one standard error.

**Figure 7 animals-13-01506-f007:**
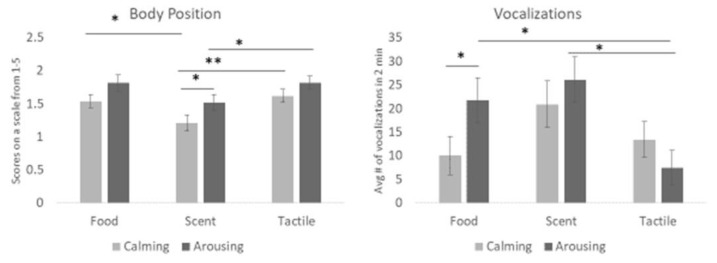
The figure illustrates the interaction between category and sensory modality. Calming scent items were associated with the lowest body position score (i.e., more relaxed) compared to the other enrichment conditions. Calming food had significantly lower vocalization frequency on average than arousing food. Arousing tactile items reduced vocalizations more than other arousing conditions. One asterisk denotes significance below the *p* < 0.05 level. Two asterisks denote significance below the *p* < 0.01 level. Error bars show ± one standard error.

**Table 1 animals-13-01506-t001:** ANOVA Tests and Means for Main Effects.

	Body Position	Vocalization
**Presence**		
Enrich M (SE)	1.67 (0.03)	16.57 (1.05)
Control M (SE)	1.74 (0.06)	24.17 (2.23)
Between F(p)	1.18 (0.278)	**9.55 (0.002)**
**Category**		
Calming M (SE)	1.47 (0.07)	14.14 (2.62)
Arousing M (SE)	1.72 (0.07)	18.45 (2.81)
Control M (SE)	1.57 (0.12)	26.63 (4.94)
Between F(p)	**3.64 (0.027)**	2.63 (0.072)
**Sensory**		
Food M (SE)	1.65 (0.08)	14.73 (3.21)
Scent M (SE)	1.37 (0.11)	23.55 (4.18)
Tactile M (SE)	1.72 (0.08)	10.46 (2.91)
Control M (SE)	1.57 (0.12)	26.63 (4.94)
Between F(p)	**2.72 (0.043)**	**4.00 (0.008)**

Note. Details the means and standard errors of all dogs in each condition. The light gray row shows the F and p for the between-group comparison of the means above the row (interaction). All comparisons significantly different at the *p* < 0.05 level are in bold.

## Data Availability

The data presented in this study are available on request from the corresponding author. The data are not publicly available due to this research being part of a larger project.

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
