# Peer review of "Ruff Morning? The Use of Environmental Enrichment during an Acute Stressor in Kenneled Shelter Dogs"

_animals, 2023, doi:10.3390/ani13091506_

Round 1

Reviewer 1 Report

This article was well-written, and easy to read; I was glad to review it.  Here are my recommendations:

13 scent of lavender: scent is not tactile; if you mean the blanket, say “lavender-scented blanket”

45    “on the rise” 7-year-old reference, do you have more recent statistics?

Figure 3 Two-dimensional pictorial measures: have these been used in other research? I found the scoring to be vague; can you describe it more clearly? Two or four if no match?  You really need to explain how your scoring was valid.

189: how far away were the observers from the dogs? Standing? Sitting? Facing?  Help the reader see what was happening

248 Your Data Analysis is hard to understand, and you are not reporting all of your outcomes. In the Results, what does the “mean” represent?

Did not mention feeding times/times exposed to human contact vs not and their effects

397 “In dogs, this stress has been shown to lead to overactivation of the 398 HPA axis and may be correlated with developing diseases such as diabetes, depression, 399 and cancer [44]. ” This article has nothing to do with your statement.

Reviewer 2 Report

Comments on the Manuscript "Ruff Morning? The Use of Environmental Enrichment During an Acute Stressor in Kenneled Shelter Dogs”, Submitted to the Animals

1.      GENERAL COMMENTS

I appreciate the opportunity to review this manuscript.

The manuscript describes a comparative study of different stimuli for domestic dogs living in a shelter. The stimulation is to increase welfare and decrease stress during cleaning procedures in the morning. The stimulus was food, tactile, and scent enrichment items to increase (arousing) or decrease (calming) the dog’s behavior. The results are suggestive that stress can be reduced using specific types of enrichment during daily distressing events.

Despite the importance of the study, it has limitations due to some out-of-context sentences and a confusing text structure. I explain my point of view in more detail below.

2.      DETAILED COMMENTS

 Line 18, Abstract

I'm afraid the abstract needs to be written if the authors agree with my comments about the methods, results, discussion, and conclusion.

Line 38: Keywords

According to the authors, daily cleaning of dog enclosures is stressful. Although the terms “acute stress” and “chronic stress” may be somewhat arbitrary, I'm hesitant to classify cleaning stress as "acute". If cleansing is done on a daily basis, how can it be considered acute after 90, 120, 300 days, etc.?

Introduction

Line 61: The statement “shelter animals have not been specifically investigated”, is not accurate. Stressful events in dog and cat shelters have been investigated in many studies (e. g. Kessler & Turner, 1999; McCobb et al, 2005; Tod et al., 2005; Coppola et al., 2006).

Line 112: The sentence “It has been found that dogs…” seems disconnected from the rest of the paragraph.

Line 118: The statement “ Despite the amount of research conducted on specific enrichment items [11; 13; 15; 21; 23; 27-32], none have compared the effectiveness of different types of enrichment”, is not accurate. There are studies comparing different stimuli for environmental enrichment for dogs, including some mentioned in this manuscript (e. g. Conley et al., 2006; Scott, et al., 2007).

Line 121: I don't understand why authors classify environmental enrichment into “animate vs. inanimate”. There does not seem to be a strong connection with the scope of the study.

Line 124: The "energy release" argument seems somewhat mechanistic and teleological. It doesn't seem to me to be a sound argument.

Line 151: see my commentary on line 118.

Material and methods

Line 174: N= 83 subjects; “There were 32 females and 50 males”. Did the total number is 83 or 82 dogs?

Change the unit of weight in pounds to Kg.

What were the standard deviations for weight and age? Write the standard deviations.

Line 228: Figure 3 is structured in numbered images that do not follow the editorial standard. Also, the resolution is poor.

Results

Table 1: Unable to understand Table 1. ANOVA test values are missing.

Line 281: the term “significance below the p = .05 level”, is not accurate. I suggest only “p< .05”. Review all similar terms in the figures.

Write in the methods that in the statistical tests, the alpha is <.05.

Discussion

Line 365: “very few studies…”, “few studies…”.

Conclusion

Lines 398 -404: In these lines, there is merely speculative information that is not linked to the results of the study.

References

The cited references are within the scope of the study.

3. REFERENCES

Conley, M.J.; Fisher, A.D.; Hemsworth, P.H. Effects of human contact and toys on the fear responses to humans of shelter-housed dogs. Applied Animal Behavior Science 2014, 156, 62-69.

Coppola, C. L., Grandin, T., & Enns, R. M. (2006). Human interaction and cortisol: can human contact reduce stress for shelter dogs?. Physiology & behavior, 87(3), 537-541.

Kessler, M. R., & Turner, D. C. (1999). Effects of Density and Cage Size on Stress in Domestic Cats (Felis Sil Vestris Catus) Housed in Animal Shelters and Boarding Catteries. Animal Welfare, 8(3), 259-267.

McCobb, E. C., Patronek, G. J., Marder, A., Dinnage, J. D., & Stone, M. S. (2005). Assessment of stress levels among cats in four animal shelters. Journal of the American Veterinary Medical Association, 226(4), 548-555.

Scott, K., Taylor, L., Gill, B. P., & Edwards, S. A. (2007). Influence of different types of environmental enrichment on the behaviour of finishing pigs in two different housing systems: 2. Ratio of pigs to enrichment. Applied Animal Behaviour Science, 105(1-3), 51-58.

Tod, E., Brander, D., & Waran, N. (2005). Efficacy of dog appeasing pheromone in reducing stress and fear related behaviour in shelter dogs. Applied Animal Behaviour Science, 93(3-4), 295-308.

Round 2

Reviewer 2 Report

Dear Authors

I appreciate the opportunity to review this manuscript again. The authors complied with most of the recommendations, substantially improving the manuscript. I think the text is more understandable. The manuscript deals with an original theme related to improvements in the situation of dogs in shelters. Therefore, the work presented here has the potential to be consulted by students, researchers, and technicians, for academic work and practical applications. In my opinion, the manuscript is acceptable for publication in the Animals.